

# Assessment of biomass potentials of microalgal communities in open pond raceways using mass cultivation

Seung-Woo Jo[1], Jeong-Mi Do[2,3], Ho Na[2,3], Ji Won Hong[4], Il-Sup Kim[5] and Ho-Sung Yoon[1,2,3,5]

[1] Department of Energy Science, Kyungpook National University, Daegu, South Korea
[2] Department of Biology, Kyungpook National University, Daegu, South Korea
[3] School of Life Sciences, BK21 Plus KNU Creative BioResearch Group, Kyungpook National University, Daegu, South Korea
[4] Department of Hydrogen and Renewable Energy, Kyungpook National University, Daegu, South Korea
[5] Advanced Bio-resource Research Center, Kyungpook National University, Daegu, South Korea

Corresponding authors
Il-Sup Kim, 92kis@hanmail.net
Ho-Sung Yoon, hsy@knu.ac.kr

## ABSTRACT

Metagenome studies have provided us with insights into the complex interactions of microorganisms with their environments and hosts. Few studies have focused on microalgae-associated metagenomes, and no study has addressed aquatic microalgae and their bacterial communities in open pond raceways (OPRs). This study explored the possibility of using microalgal biomasses from OPRs for biodiesel and biofertilizer production. The fatty acid profiles of the biomasses and the physical and chemical properties of derived fuels were evaluated. In addition, the phenotype-based environmental adaptation ability of soybean plants was assessed. The growth rate, biomass, and lipid productivity of microalgae were also examined during mass cultivation from April to November 2017. Metagenomics analysis using MiSeq identified ~127 eukaryotic phylotypes following mass cultivation with (OPR 1) or without (OPR 3) a semitransparent film. Of these, ~80 phylotypes were found in both OPRs, while 23 and 24 phylotypes were identified in OPRs 1 and 3, respectively. The phylotypes belonged to various genera, such as *Desmodesmus*, *Pseudopediastrum*, *Tetradesmus*, and *Chlorella*, of which, the dominant microalgal species was *Desmodesmus* sp. On average, OPRs 1 and 3 produced ~8.6 and 9.9 g m$^{-2}$ d$^{-1}$ (0.307 and 0.309 DW L$^{-1}$) of total biomass, respectively, of which 14.0 and 13.3 wt% respectively, was lipid content. Fatty acid profiling revealed that total saturated fatty acids (mainly C16:0) of biodiesel obtained from the microalgal biomasses in OPRs 1 and 3 were 34.93% and 32.85%, respectively; total monounsaturated fatty acids (C16:1 and C18:1) were 32.40% and 31.64%, respectively; and polyunsaturated fatty acids (including C18:3) were 32.68% and 35.50%, respectively. Fuel properties determined by empirical equations were within the limits of biodiesel standards ASTM D6751 and EN 14214. Culture solutions with or without microalgal biomasses enhanced the environmental adaptation ability of soybean plants, increasing their seed production. Therefore, microalgal biomass produced through mass cultivation is excellent feedstock for producing high-quality biodiesel and biofertilizer.

## INTRODUCTION

Microalgae, which have evolved over a long period, are biologically diverse organisms that are highly valuable for academic and industrial fields. Microalgae have high cellular lipid content and so have distinct characteristics that may be used for biodiesel production or for the production of various other useful substances, such as valuable unsaturated fatty acids (UFAs), proteins, vitamins, and carotenoid secondary metabolites (*Ren et al., 2017*). Therefore, microalgae are increasingly used to produce high-value products and bioactive substances in a wide spectrum of fields ranging from sustainable energy to healthcare (*Jebali et al., 2019*). Global microalgal research can be considered the next-generation industry that will spearhead the growth of a green and low-carbon era (*Bell et al., 2016*).

Microalgae can generate large amounts of carbon-rich fatty acids from $CO_2$ and grow fast in poor-quality waters or on non-arable lands without depleting food resources, so they are effective biofuel sources (*Simas-Rodrigues et al., 2015*; *Faried et al., 2017*; *Ren et al., 2017*). Microalgal strains of industrial value are found by either characterizing natural strains or establishing artificial strains through genetic manipulation. Isolating of new microalgal strains from the environment enables the finding of new strains that have unique physiological features and are promising biofuel sources (*Abou-Shanab et al., 2011*). This approach is effective because of myriads of uncharacterized microalgal strains in nature. However, artificial strain improvement through genetic engineering depends exclusively on the microalgal genetic information available (*Cheng et al., 2019*). Unfortunately, despite the industrial importance of microalgae, research on microalgal genetic engineering is limited compared to similar research on other organisms (*Ng et al., 2017*). Microalgal genetic engineering technologies have been fully established only for *Chlamydomonas reinhardtii* (*Young & Purton, 2016*). Although other microalgal strains have been successfully transformed, these are usually one-time cases with low reproducibility (*Siddiqui et al., 2019*). In addition, native isolates are well adapted to local conditions and exhibit better performance and robustness (*Alishah Aratboni et al., 2019*). Therefore, for large-scale industrial uses of microalgae, it is necessary to establish a microalgal mass cultivation system, such as the open pond raceway (OPR), using indigenous microalgal species instead of engineered strains.

To realize the potential of microalgae, improvements are required in cost, productivity, and sustainability (*Dahlin et al., 2018*). To scale up production at lower costs, open pond systems coupled with fresh water or wastewater are a viable alternative (*Bell et al., 2016*). These systems are operated in semi-continuous or continuous mode to meet industrial production standards by maximizing both biomass and lipid productivities, while ensuring the industrial suitability of the microalga-derived fatty acids (*Song et al., 2013*). However, although outdoor productivity is often less than the laboratory yield because of the extreme dynamics of environmental conditions, such as water temperature and light intensity, studies on identifying optimal strains have rarely investigated these criteria in semi-continuous or continuous cultures (*Dahlin et al., 2018*). Recently, several studies questioned the sustainability of biofuel production from freshwater species because it is uneconomical (*Chisti, 2013*). Because of these limitations, most studies on microalgae are

conducted in closed photobioreactors (*Ren et al., 2017*). However, an open pond system is still needed in order to produce bioenergy as an alternative to fossil fuel.

In industrial-scale open pond cultivation with fresh water, many environmental (light and temperature), operational (pH, $CO_2$, and nutrients), and biological (bacteria and fungi) parameters affect microalgal diversity and productivity, in turn affecting biomass quality and quantity (*Cho et al., 2015*). Therefore, open pond systems are prone to colonization by environmental microbes and may contain numerous distinct taxa (*El-Sheekh et al., 2019*). In the microalgal-based biofuel field, maintaining the integrity of the dominant microalgal species is mandatory, so there is a lot of uncertainty around biofuel production through freshwater systems (*Roccuzzo, Beckerman & Pandhal, 2016*). In addition, the feasibility of maintaining dominant microalgal communities in fresh water-dependent ponds and the factors involved thereof are unclear (*Park, Craggs & Shilton, 2013*). Undesirable colonization can be controlled by using locally adapted microalgae observed in natural ecosystems, and extensive studies have been conducted to probe locally isolated microalgae strains for their potential to be used in biofuel production (*Rodolfi et al., 2009*).

Metagenome technologies are powerful tools for analyzing complex microbial communities and have led to a tremendous increase in the knowledge of the functions, protein families, biotechnology, and ecology of microbial communities (*Hugenholtz et al., 1998*; *Krohn-Molt et al., 2017*). Metagenome studies have provided us with insight into the complex interactions of microorganisms with their environments and hosts (*Krohn-Molt et al., 2017*). Surprisingly, few studies have focused on microalgae-associated metagenomes (*Krohn-Molt et al., 2017*; *Sambles et al., 2017*), and no study has addressed aquatic microalgae and their bacterial communities in OPRs. However, the microalgal community based on phylogenetic and metagenomic diversity is growing.

This study investigated the effect of various factors on the diversity and productivity of an indigenous microalgal community in OPRs with (OPR 1) or without (OPR 3) a semitransparent film through three seasons. We modeled the biodiversity of a microalgal mass cultivation system by considering environmental factors, such as water temperature, intra- and interspecies competition in the microalgal community, and the abundance of other species, including cyanobacteria. We measured biomass-based lipid production in fresh water dependent OPRs operated during mass cultivation from April to November 2017. To identify locally adapted microalgal strains, we examined and statistically analyzed various environmental (meteorological and nutritional), biological (algal and cyanobacterial diversity), and physiochemical (fatty acid profile and biodiesel quality) parameters. Our findings may facilitate biofuel and biofertilizer production from freshwater system biomasses.

## MATERIAL AND METHODS

### Operation of an OPR system

A microalgal semi-continuous mass cultivation system comprising two OPRs ($\sim$675.0 $m^2$) with (OPR 1) or without (OPR 3) a semitransparent film at Chilgok-gun Agricultural Technology Center (36°02′18.91N″, 128°22′57.7E″), Korea, was operated to monitor

the ecological system and biodiversity in microalgae-dependent mode under natural environmental conditions. The mass cultivation period was from April 1 to November 30, 2017. OPRs used natural sunlight. We supplemented 236 tons of groundwater with Eco-Sol (nitrogen:phosphorus:potassium = 25:9:18; Farm Hannong, Ulsan, Korea) as a commercial water-soluble fertilizer to produce a final concentration of 10.0–18.0 mg L$^{-1}$ total nitrogen (TN) and 2.0–3.0 mg L$^{-1}$ total phosphorus (TP). Next, we injected 10% of $CO_2$ in air at a flow rate of 10 L min$^{-1}$ in both OPRs using a venturi system during daylight hours (*Hong et al., 2017*). The OPR temperature and pH were measured every 3 h using a WQC-24 portable multiparameter water quality meter (Multi Measuring Instruments Co. Ltd., Tokyo, Japan).

## Microalgae mass cultivation

Indigenous microalgae from earlier cultivation periods were used. Briefly, *Desmodesmus* sp., the dominant species grown in OPRs, was inoculated, and species succession was monitored at morphological and molecular levels. Semi-continuous cultivation was conducted at a velocity of 25–30 cm s$^{-1}$. Next, we harvested 155 tons (approximately two-thirds of the microalgal culture from each raceway) and replaced it with the same volume of underground water with appropriate nutrient levels; the remaining culture was used as a seed. The mass cultivation was continuously recycled throughout the study. When TN and TP were almost depleted and the optical density (OD) and total biomass reached their maximum values, the culture was harvested. TN and TP levels were quantitated using HS-TN(CA)-L and HS-TP-L water test kits (Humas, Daejeon, Korea), and OD was measured using a spectrophotometer (X-ma 1200V; Human Corp., Seoul, Korea) at a wavelength of at 680 nm. The total biomass was dried in a dry oven and weighed. Finally, to observe morphological changes, the microalgal cultures were sampled every 3 days and inspected at 400 × magnification under an Eclipse E100 microscope (Nikon Instruments Inc., Tokyo, Japan).

## MiSeq-based microalgal diversity

To determine mock communities, we screened OPRs for the presence of *Acutodesmus* sp. KNUA038, *Chlamydomonas* sp. KNUA021, *Chlamydomonas* sp. KNUA040, *Chlorella* sp. KNUA027, *Desmodesmus* sp. KNUA024, *Pseudopediastrum* sp. KNUA039, and *Scenedesmus obliquus* KNUA019 by sequencing 18S ribosomal RNA(rRNA) V4 and V8–V9 regions, as previously described (*Bradley, Pinto & Guest, 2016*). We also confirmed microalgal identification by using the Basic Local Alignment Search Tool (BLAST) against the National Center for Biotechnology Information (NCBI) GenBank database.

Next, we extracted genomic DNA (gDNA) from the harvested biomass using a DNeasy Plant Mini kit (Qiagen, Hilden, Germany) and purified it using a Wizard DNA Clean-Up System (Promega, Madison, WI, USA) according to the manufacturers' instructions. Then, 16S and 18S rRNA genes were used to identify cyanobacteria and microalgae, respectively, within the OPRs. Each sequencing sample was prepared in accordance with Illumina 16S and 18S metagenomic sequencing library protocols, as previously described (*de Muinck et al., 2017*; *Kim et al., 2017*). Briefly, for 16S rRNA, the internal transcribed spacer 2 (ITS2)

region was amplified using PCR with the ITS3/ITS4 primer set (*de Muinck et al., 2017*; *Kim et al., 2017*). The 18S protocol was designed to broadly recognize eukaryotes, favoring microbial eukaryotic lineages. The primers were based on a previous report (*Amaral-Zettler et al., 2009*) and designed to be used with the Illumina MiSeq platform ((*Bradley, Pinto & Guest, 2016*). The 18S libraries on MiSeq were run, as previously described (*Caporaso et al., 2012*), and PCR amplicons derived from the conserved V8–V9 region by using the Mi1422F/Mi1510R primer pair were sequenced. Table S1 shows the primer sets overhanging the pre-adapter and sequencing adapter. The outlines of the 16S and 18S protocols were the same. Sequences were processed and data mining performed, as previously described (*Kozich et al., 2013*; *Bradley, Pinto & Guest, 2016*). Finally, the consensus taxonomy of the operational taxonomic unit (OTU) was constructed using Silva v119 taxonomy information (*Ondov, Bergman & Phillippy, 2011*).

## Microalgal biomass characterization

First, the harvested biomass was freeze-dried, pulverized using a mortar and pestle, and sieved through an ASTM No. 230 mesh (pore size = 63 $\mu$m). Next, the total lipid content was determined using a the sulfo-phospho-vanillin colorimetric method, as previously described with modifications (*Mishra et al., 2014*). Proximate analysis was performed to measure ash content using a DTG-60A thermal analyzer (Shimadzu, Kyoto, Japan) and platinum pans containing ~10 mg of each sample or 30 mg of $\alpha$-alumina ($\alpha$-Al$_2$O$_3$) powder (Shimadzu) as a reference. Nitrogen (>99.999%, N$_2$) was supplied as the carrier gas at a rate of 25 mL min$^{-1}$ to prevent oxidation of the microalgae powder. The samples were heated from 50 to 900 °C at a rate of 10 °C min$^{-1}$. Thermogravimetric analysis (TGA) data were obtained using ta60 ver. 2.21 software (Shimadzu) according to the manufacturer's instructions. Ultimate analysis was performed using a Flash 2000 elemental analyzer (Thermo Fisher Scientific, Milan, Italy) to determine the carbon (C), hydrogen (H), nitrogen (N), and sulfur (S) content, and the oxygen (O) content was calculated by subtracting the ash and CHNS content from the total. The calorific value (CV, also known as higher heating value [HHV]) was estimated using the following equation developed by *Friedl et al. (2005)*:

$$CV = \left\{ 3.55C^2 - 232C - 2,230H + 51.2C \times H + 131N + 20,600 \, (MJ/kg) \right\}.$$

## Gas chromatography-mass spectrometry analysis

Lipid extraction was performed as previously described (*Breuer et al., 2013*). The fatty acid methyl ester (FAME) composition was analyzed using a 7890A gas chromatograph equipped with a 5975C mass selective detector (Agilent, Santa Clara, CA, USA). Gas chromatography (GC) runs were performed using a DB-FFAP column (30 m, 250 $\mu$m ID, 0.25 $\mu$m film thickness; Agilent). The initial oven temperature of the gas chromatograph was set to 50 °C and maintained for 1 min; subsequently, it was increased first to 200 °C at a rate of 10 °C min$^{-1}$ for 30 min and then to 240 °C at a rate of 10 °C min$^{-1}$, which was maintained for 20 min. The injection volume was 1 $\mu$L, with a split ratio of 20: 1. Helium

was used as a carrier gas at a constant flow rate of 1 mL min$^{-1}$. The mass spectrometry (MS) parameters were as follows: injector and source temperatures of 250 and 230 °C, respectively; electron impact mode at an acceleration voltage of 70 eV for sample ionization; and acquisition range of 50–550 $m/z$. Compound identification was performed by matching the mass spectra with those in Wiley/NBS libraries; matches >90% were considered valid.

## Biodiesel quality assessment

The following physical parameters were measured to assess biodiesel quality: saponification value (SV), iodine value (IV), degree of unsaturation (DU), cetane number (CN), long-chain saturated factor (LCSF), and cold filter plugging point (CFPP). The parameters were quantitated on the basis of the fatty acid compositions using the following empirical equations, as previously described (*Ramos et al., 2009*):

$$SV = \sum \frac{560 \times N}{M}$$

$$IV = \sum \frac{254 \times N \times D}{M}$$

$$CN = \left(46.3 + \frac{5458}{SV}\right) - (0.225 \times IV)$$

$$LCSF = \{0.1 \times N(C16:0)\} + \{0.5 \times N(C18:0)\} + \{1 \times N(C20:0)\} + \{1.5 \times N(C22:0)\}$$
$$+ \{2 \times N(C24:0)\}$$

$$CFPP = (3.1417 \times LCSF) - 16.477$$

$$DU = MUFA + (2 \times PUFA)$$

where $D$, $M$, and $N$ denote the number of double bonds, average molecular mass, and weight percent (wt%) of each fatty acid, respectively; MUFA refers to monounsaturated fatty acid; and PUFA refers to polyunsaturated fatty acid. C16:0, C18:0, C20:0, and C22:0 are also in wt%.

## Biological enhancer assay in soybean plants

First, soybean seeds (Daewon) were sown in cultivation soil, germinated at 30 °C for ~7 days in a growth chamber (70–100 μmol m$^{-2}$ s$^{-1}$) in a 14 h/10 h dark cycle, at 60% humidity, and grown to the V2 stage in the same incubator. Next, 15 soybean seedlings were transplanted to a large pot 30 cm in diameter and then cultivated near a mass culture open pond system. Subsequently, we applied the following treatments to the soybeans, with the solutions applied every 3 days:

1. The control pot was supplemented with 1 L of groundwater.
2. The experimental pot was supplemented with 1 L of the eco-sol medium as a soil drench.
3. The supernatant (1 L) of the microalgal culture was collected by centrifugation at 4,000 rpm for 20 min at 20 °C and mixed with the eco-sol medium as a soil drench.
4. The culture solution (1 L) containing microalgal biomass and the supernatant as a soil drench.
5. The pot with soybean plants was sprayed with 1 L of microalga-free supernatant.

The soybean plants' environmental adaptation was assessed by monitoring the plant phenotype, and soybean productivity was measured by examining seed maturation and the pod weight per seedling.

## Statistical analysis

All experiments were performed in at least triplicate. Data were shown as the average of triplicates, and error bars represented the standard deviation. $P < 0.05$ was considered statistically significant.

## RESULTS

### Biodiversity of mass cultivation in open pond raceways
#### Establishment of MiSeq-based qualitative verification

First, we examined the reported primer sets to determine whether the microalgal communitiy comprising the seven known microalgal species (*Acutodesmus* sp. KNUA038, *Chlamydomonas* sp. KNUA021, *Chlamydomonas* sp. KNUA040, *Chlorella* sp. KNUA027, *Desmodesmus* sp. KNUA024, *Pseudopediastrum* sp. KNUA039, and *Scenedesmus obliquus* KNUA019) could be effectively characterized by high-throughput sequencing (HTS). PCR amplification with primers for actin, *rbcL*, and 18S rRNA V4, V8–V9, and V9 regions yielded 180–650 bp amplicons from the pooled gDNA isolated analyzed samples (Fig. S1, and Data S1). We obtained an amplicon of ∼610 bp with *rbcL* primers 28S2F/28S2R (Fig. S1B). However, no amplicon was detected in analyzed samples with actin primers (Fig. S1A).

Next, we introduced various regions of the 18S rRNA gene into HTS-based comparative analysis. Of them, V4 and V8–V9 regions were significant, and the taxonomic groups for the V8–V9 region showed higher concordance rates compared to the V4 region. The V4 region failed to reliably capture two of the seven microalgal species included in this study (*Acutodesmus* sp. KNUA0038 [accession no. KT883908] and *Scenedesmus* sp. KNUA019 [accession no. MT644350]), while the V8–V9 region accurately represented the mean relative abundance (Tables S2 and S3). However, the V9 region was too short (<200 nucleotides [nt]) to allow overlap between forward and reverse reads in the Illumina Miseq platform (250–300-nt single read length, resulting in 450–500-nt-long combined reads with 50–150 bp overlaps) (*Bradley, Pinto & Guest, 2016*). Results showed that the V8–V9 region of the 18S rRNA gene is more effective for HTS compared to other candidate genes because 400–500 bp amplicons are required. Figure S2A shows schematic diagrams showing the biodiversity analysis.

#### Changes in environmental factors and growth parameter during open-pond mass cultivation

We analyzed environmental conditions in OPRs 1 and 3 during mass cultivation from April to November 2017 because the water surface was frozen in OPR 3 (without a semitransparent film) from December 2017 to February 2018 but not in OPR 1 (with the same semitransparent film) (Fig. S2B and Data S2). The mean water temperature in spring (April–May), summer (June–August), and the fall (September–November), was 17.4, 25.4,

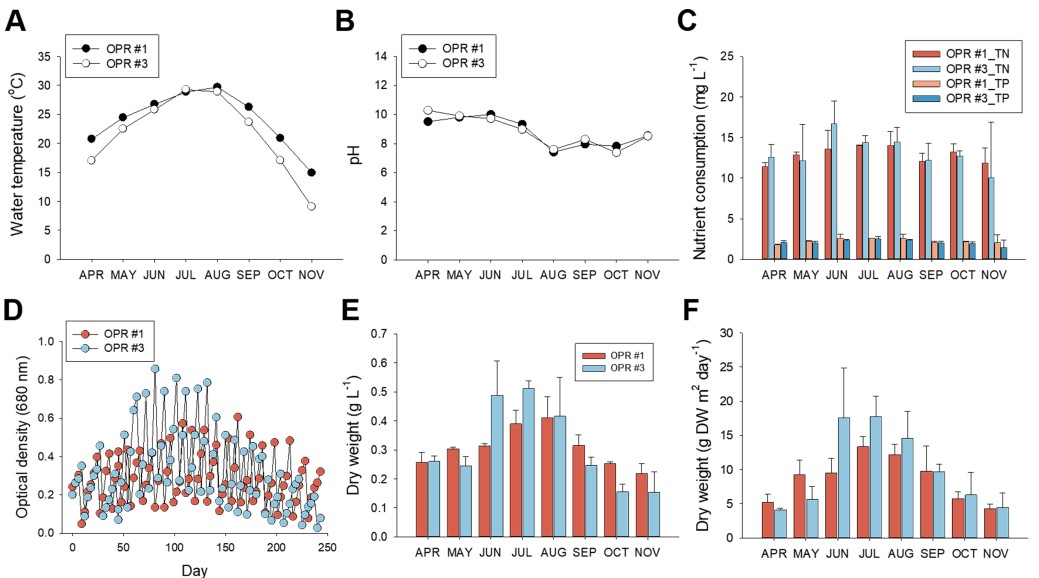

**Figure 1  Nutrient consumption, biomass levels, and lipid content in relation to cultivation parameters from April to November, 2017.** Environmental cultivation factors were monitored by measuring the (A) water temperature and (B) pH. Black circle, OPR 1; white circle, OPR 3. (C) Nutrient consumption rate assessed by examining removal of TN and TP. Red, OPR 1 TN; light-blue, OPR 3 TN; light-red, OPR 1 TP; blue, OPR 3 TP. (D) Biomass monitored by measuring the OD at 680 nm for the indicated time. Light-red circle, OPR 1; light-blue circle, OPR 3. (E, F) Biomass productivity also represented DW (g·L$^{-1}$ and g DW m$^{-2}$·day$^{-1}$). Light-red bar, OPR 1; light-blue bar, OPR 3. OPR, open pond raceway; TN, total nitrate; TP, total phosphorous; OD, optical density; DW, dry weight.

and 14.1 °C, respectively (Fig. 1A). Although we observed no differences between OPRs 1 and 3, the temperature of OPR 1 was higher compared to OPR 3 during mass cultivation, except for summer. In addition, pH was neutral to heavy alkaline (7.2–10.4), becoming increasingly neutral with a gradual increase in water temperature (Fig. 1B).

With regard to dissolved nutrients, TN and TP levels were strongly depleted in summer compared to spring and the fall, concurrent with a rapid increase in total biomass and lipid levels. Compared to initial TN levels (∼17 mg L$^{-1}$), ∼13.57–16.69 mg L$^{-1}$ was removed in summer and ∼10.02–13.21 mg L$^{-1}$ in spring and the fall. Initial TP levels were 2.45–2.63 mg L$^{-1}$, 2.35–2.62 mg L$^{-1}$ in summer, and 1.47–2.22 mg L$^{-1}$ in spring and the fall (Fig. 1C, and Data S2). Overall, TN and TP consumption was 12.87 mg L$^{-1}$ (86.2%) and 2.26 mg L$^{-1}$ (96.3%), respectively, in OPR 1 but 13.14 mg L$^{-1}$ (90.1%) and 2.10 mg L$^{-1}$ (93.5%) in OPR 3, which confirmed the higher nutrient removal rate in TP compared to TN.

We also investigated the relationship between nutrient consumption and biomass productivity. Total biomass was based on growth kinetics measured from the OD and DW. The absorbance and DW of OPR #3 were higher compared to OPR 1 in June–August 2017, with the highest difference in summer, when water temperature increased. OPR 3 had a maximum biomass production of 17.6 g m$^{-2}$ d$^{-1}$, while OPR 1 in the greenhouse had a maximum microalgal biomass production of 13.3 g m$^{-2}$ d$^{-1}$ (Fig. 1E and Data S2).

OPRs 1 and 3 had a mean total microalgal biomass production of ~8.6–9.9 g m$^{-2}$ d$^{-1}$, corresponding to 0.307–0.309 DW L$^{-1}$ (Figs. 1D–1F and Data S2).

### Overview of MiSeq-based microalgal community during mass cultivation

We monitored the microalgal community using MiSeq with DNA isolated from the pooled microalgal biomass during mass cultivation from April to November 2017. PCR products were visualized to identify contaminated samples (Fig. S2C) and then used in MiSeq. MiSeq analysis produced results for the OTU and Shannon ($H'$) and Simpson ($D'$) indices (Data S3). The number of OTUs in OPR 1 and 3 significantly increased in the fall compared to spring and summer, although the two OPRs showed a slight difference. The $H'$ index in July–November 2017 was within the typical range of 1.5–3.5, as previously reported (*Bradley, Pinto & Guest, 2016*), and in April–June 2017 was rarely <1.5. During the testing period, the $D$ index, the most common dominance parameter, decreased with time, indicating that diversity increases with regard to evenness (Fig. S3). Richness and evenness are confounded in the $H'$ index, so many biodiversity researchers prefer to use two indices for comparative studies, combining a direct estimate of species richness (i.e., the total number of species in the community) with some measurement of dominance or evenness (*Caporaso et al., 2010*).

### Microalgal succession during mass cultivation

The following eukaryotic organisms, including microalgae, were identified in OPR 1: *Desmodesmus* sp. (62%), *Heterocypris* sp. (5%), *Amoeboaphelidium* sp. (4%), *Pseudodifflugia* cf. (2%), *Chlorella* sp. (1%), *Coelastrum pseudomicroporum* (1%), *Pseudopediastrum integrum* (1%), *Chlamydomonadales* sp. (0.9%), *Tetradesmus obliquus* (0.8%), *Hariotina reticulate* (0.3%), *Scenedesmus* sp. (0.1%), and others (18%), such as uncultured eukaryotes (Fig. 2). The following species were identified in OPR 3: *Desmodesmus* sp. (74%), *T. obliquus* (1%), and others (22%) (Fig. 2 and Fig. S4, and Data S3 and S4). Although a greater number of microalgal species were found in OPR 1, the highly dominant microalgal species in both OPRs 1 and 3 was *Desmodesmus* sp. (Fig. 2). The *Desmodesmus* sp. population of OPR 3 was 12% higher than that of OPR 1.

As water temperature in OPR 1 and 3 increased in spring (Fig. 1A), some green microalgae (e.g., *Desmodesmus* sp. and *T. obliquus*) showed a change in their taxonomic ranks (Fig. 3). The microalgal bloom period in the spring was ~3–4 weeks because of competition within the cultivation tank, and the effect of increased numbers of other eukaryotic organisms switched every 2–3 weeks (Fig. S4). In late spring, green microalgae (Chlorophyta), such as *Desmodesmus* sp., proliferated rapidly as water temperature increased, becoming the dominant species (Fig. S7A–S7D). In summer, various species, including green microalgae, such as *Desmodesmus* sp., *Chlorella* sp., *C. pseudomicroporum*, and *P. integrum*, proliferated more vigorously in OPR 1 compared to OPR 3 (Fig. 3, Figs. S5A–S5E, and S8). In addition, the proportion of *Heterocypris* species, known as *Ostracoda*, increased to 33% in OPR 1 in summer (August) before disappearing after the fall when water temperature decreased (Figs. S1A and S9A, and Data S4). In OPR 3, *Nucleariidae* was abundant in summer but disappeared after the fall (Fig. 3 and Fig. S9B). In the fall, various species that were highly abundant in summer disappeared because of low water temperatures (Fig. 1A and

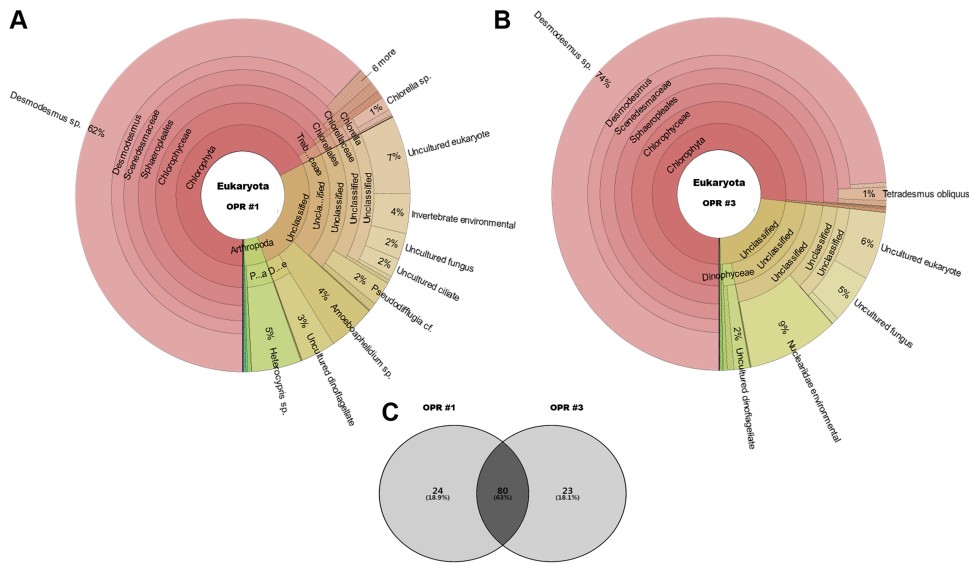

**Figure 2** **Krona graphs showing the average population of eukaryotic community.** MiSeq-based identification of the eukaryotic community, including microalgae, from April to November during mass cultivation in (A) OPR 1 and (B) OPR 3. (C) The eukaryotic community was described at the species-level, with ~80 species found in both OPRs 1 and 3, 24 in OPR 1, and 23 in OPR 3. OPR, open pond raceway.

Figs. S9C–S9D, and S10). *Desmodesmus* sp. became dominant again in OPRs 1 and 3, and the proportions of *Coelastrum* sp., *Golenkinia longispicula*, and *Micractinium* sp. increased in OPR 1. *Chlorella miniata* appeared in the fall as well (Fig. 3 and S10A). Overall, our results indicated that the difference between ecologically stabilized (OPR 3) and artificial (OPR 1) cultivation systems can be distinguished by their microalgal flora contents and that *Desmodesmus* sp. is a potential species for mass cultivation, especially in an open pond system.

### Cyanobacterial community during mass cultivation

We identified 26 cyanobacteria species (Figs. S11 and S12, and Data S5 and S6). Of these, *Cyanobium gracile* frequently appeared in both OPR 1 and 3 in spring and summer, while *Foliisarcina bertiogensis*, *Spirulina major*, *Cyanobacterium aponinum*, and *Microcystis aeruginosa* bloomed in the fall, but their populations were higher in OPR 1 compared to OPR 3, and *Synechococcus elongatus* and *Calothrix desertica* were identified only in OPR 3 in the fall (Fig. S12F–S12H). The relative abundance of other cyanobacterial species identified was <1% (Fig. S14). Overall, the number of emerging microalgal species in OPRs 1 and 3 was high in summer and low in spring and the fall, consistent with cyanobacterial species emergence, which are highly abundant in summer. In addition, three dinoflagellate and one cryptomonad species were also identified. In wastewater treatment coupled with biofuel applications, cyanobacteria continue to dominate during the fall and winter, which has an overarching influence on microalgal diversity and microalgal biomass (*Cho et al., 2015*). Therefore, we need a good understanding of beneficial interactions between microalgae and

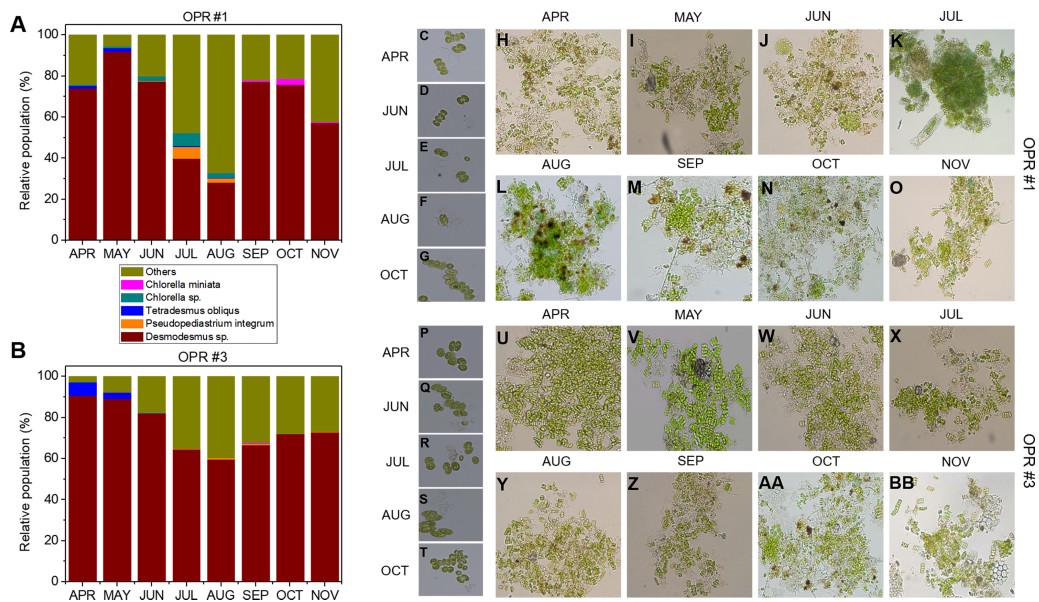

**Figure 3 Microalgal community from April to November during the mass cultivation period.** Population abundance of microalgal species in (A) OPR 1 and (B) OPR 3. Wine, *Desmodesmus* sp.; orange, *Pseudopediastrum integrum*; blue, *Tetradesmus obliquus*; dark cyan, *Chlorella* sp.; magenta, *Chlorella miniata*; dark yellow, others. Microalgal calendar in OPRs 1 and 3 during mass cultivation. Dominant microalgal strain in (C–G) OPR 1 and (P–T) OPR 3 from April to November 2017. Microscopy-based microalgal richness in (H–O) OPR 1 and (U–BB) OPR 3 during the same period. OPR, open pond raceway.

other organisms, including cyanobacteria, for exploring the biotechnological applications of microalgae-based biomasses.

## Proximate and ultimate analyses

A higher percentage of bioenergy parameters such as volatile matter (VM; >80%) is advantageous for biofuel production and biodiesel quality, so we performed quantitative estimation of the microalgal biomass using proximate and ultimate analyses (*Awaluddin et al., 2016*). We identified very small differences between OPRs 1 and 3 during culture. In proximate results, moisture, VM, and ash contents were 4.2–8.3, 71.5–84.0, and 10.5–21.7 wt%, respectively. In addition, moisture levels were slightly higher in spring and the fall in OPR 1, although it was difficult to determine a regular pattern for moisture in OPRs 1 and 3. The increased moisture value in spring gradually decreased in summer and again increased in the fall, which was observed in both OPRs 1 and 3. The VM content was the highest in summer and the fall, with OPR 1 being ~2–3% higher than OPR 3, while the ash content reached the highest level in spring, with OPR 3 being ~4–5% higher than OPR 1. Overall, the mean weight percentage values of moisture, VM, and ash content were 6.68, 79.73, and 13.57, respectively, in OPR 1 and 6.10, 77.42, and 16.48, respectively, in OPR 3 (Table 1).

In ultimate, carbon, nitrogen, and hydrogen levels were relatively higher in summer and the fall relative to levels in spring and higher in OPR 1 compared to OPR 3, consistent with the increased water temperature during culture. The sulfur and oxygen content were

**Table 1 Proximate and ultimate analyses of the harvested biomass during mass cultivation.**

| | APR | | MAY | | JUN | | JUL | | AUG | | SEP | | OCT | | NOV | |
|---|---|---|---|---|---|---|---|---|---|---|---|---|---|---|---|---|
| | O1 | O3 | O1 | O3 | O1 | O3 | O1 | O3 | O1 | O3 | O1 | O3 | O1 | O3 | O1 | O3 |
| Proximate analysis (wt %) | | | | | | | | | | | | | | | | |
| Moisture | 7.4 ± 0.4 | 6.3 ± 0.0 | 8.3 ± 0.2 | 6.9 ± 0.4 | 6.3 ± 0.2 | 7.0 ± 0.1 | 5.9 ± 0.1 | 4.2 ± 0.3 | 5.0 ± 0.0 | 6.9 ± 0.0 | 5.2 ± 0.2 | 6.8 ± 0.2 | 8.3 ± 0.2 | 6.2 ± 0.2 | 7.1 ± 0.1 | 4.5 ± 0.1 |
| Volatile matter | 75.6 ± 0.6 | 73.6 ± 0.4 | 76.1 ± 0.2 | 71.5 ± 0.1 | 80.3 ± 0.3 | 76.8 ± 0.6 | 77.8 ± 0.3 | 78.5 ± 0.4 | 84.0 ± 0.1 | 80.3 ± 0.1 | 83.2 ± 0.1 | 81.2 ± 0.4 | 81.2 ± 0.0 | 80.8 ± 0.1 | 79.7 ± 0.1 | 76.7 ± 0.1 |
| Ash | 17.0 ± 0.2 | 20.1 ± 0.4 | 15.6 ± 0.4 | 21.7 ± 0.5 | 13.4 ± 0.6 | 16.3 ± 0.5 | 16.2 ± 0.2 | 17.3 ± 0.2 | 11.0 ± 0.1 | 12.8 ± 0.1 | 11.7 ± 0.3 | 12.0 ± 0.2 | 10.5 ± 0.2 | 12.9 ± 0.1 | 13.2 ± 0.3 | 18.8 ± 0.1 |
| Ultimate analysis (wt %) | | | | | | | | | | | | | | | | |
| Carbon (C) | 39.9 ± 0.4 | 38.6 ± 0.4 | 41.7 ± 0.0 | 39.1 ± 0.7 | 43.6 ± 0.2 | 40.8 ± 0.2 | 42.5 ± 0.1 | 41.8 ± 0.8 | 45.8 ± 0.6 | 45.7 ± 0.2 | 42.7 ± 0.2 | 43.0 ± 0.2 | 46.1 ± 0.0 | 45.1 ± 0.2 | 43.5 ± 0.2 | 37.9 ± 0.5 |
| Hydrogen (H) | 5.4 ± 0.1 | 5.4 ± 0.1 | 6.0 ± 0.0 | 5.3 ± 0.1 | 6.3 ± 0.1 | 5.7 ± 0.0 | 6.2 ± 0.0 | 5.9 ± 0.1 | 6.6 ± 0.1 | 6.4 ± 0.0 | 6.5 ± 0.0 | 6.5 ± 0.1 | 6.5 ± 0.1 | 6.4 ± 0.1 | 6.4 ± 0.1 | 5.3 ± 0.1 |
| Nitrogen (N) | 5.4 ± 0.1 | 5.8 ± 0.2 | 6.2 ± 0.1 | 5.5 ± 0.2 | 7.0 ± 0.0 | 5.4 ± 0.2 | 5.7 ± 0.1 | 5.8 ± 0.1 | 5.3 ± 0.1 | 6.9 ± 0.0 | 6.0 ± 0.0 | 6.0 ± 0.0 | 7.2 ± 0.1 | 5.6 ± 0.1 | 7.1 ± 0.0 | 4.0 ± 0.1 |
| Sulfur (S) | 0.4 ± 0.0 | 0.5 ± 0.0 | 0.4 ± 0.0 | 0.4 ± 0.0 | 0.5 ± 0.0 | 0.4 ± 0.0 | 0.4 ± 0.0 | 0.4 ± 0.0 | 0.5 ± 0.0 | 0.5 ± 0.0 | 0.4 ± 0.0 | 0.4 ± 0.0 | 0.6 ± 0.0 | 0.4 ± 0.0 | 0.5 ± 0.0 | 0.3 ± 0.0 |
| Oxygen (O) (by difference) | 31.8 ± 0.4 | 29.5 ± 0.2 | 30.2 ± 0.3 | 28.0 ± 1.5 | 29.3 ± 0.4 | 31.6 ± 0.1 | 28.9 ± 0.2 | 28.8 ± 1.1 | 30.9 ± 0.9 | 27.7 ± 0.0 | 32.8 ± 0.2 | 32.2 ± 0.2 | 29.2 ± 0.3 | 20.5 ± 0.2 | 29.4 ± 0.0 | 15.9 ± 0.4 |
| HHV (MJ/kg) | 17.0 ± 0.3 | 17.0 ± 0.2 | 18.6 ± 0.1 | 17.2 ± 0.6 | 19.8 ± 0.1 | 17.7 ± 0.1 | 19.5 ± 0.0 | 18.8 ± 0.6 | 20.8 ± 0.5 | 20.9 ± 0.1 | 19.4 ± 0.0 | 19.5 ± 0.3 | 21.0 ± 0.1 | 20.5 ± 0.2 | 20.0 ± 0.2 | 15.9 ± 0.4 |

**Notes.**

O1, Open pond raceway #1; O3, Open pond raceway #3.

also higher in OPR 1 compared to OPR 3. The mean weight percentage values of carbon, hydrogen, nitrogen, sulfur, and oxygen (CHNSO) were 43.22, 6.23, 6.23, 0.46, and 30.31, respectively, in OPR 1 and 41.50, 5.86, 5.62, 0.41, and 26.77, respectively, in OPR 3. The CV was higher in summer and the fall compared to spring and higher in OPR 1 compared to OPR 3 (Table 1). The mean CV of OPR 1 and 3 was 19.51 and 18.43 MJ kg$^{-1}$, respectively, which are higher compared to the land plant biomass (18.4 MJ kg$^{-1}$) (*Ross et al., 2008*). Therefore, the microalgal biomass grown in an open pond using mass cultivation, especially OPR 1, can have the potential to be used for producing biofuels of the desired quality.

## Fatty acid profiles

Figure 4 shows the fatty acid compositions of the microalgal biomasses in OPRs 1 and 3 during mass cultivation from April to November 2017. The major fatty acids analyzed were myristic (C14:0), palmitic (C16:0), palmitoleic (C16:1), palmitidonic (C16:4), oleic (C18:1), linolelaidic (C18:2), and linolenic (C18:3) acids. C16:0 was the most abundant fatty acid in both OPRs throughout the culture period (21.9–36.5%), while C18:1 was the second-most abundant fatty acid in OPR 1, which increased from spring to summer when water temperature was high and then decreased in the fall. In contrast, C18:1 increased in OPR 3 in spring and the fall and slightly decreased in summer (range 10.1–28.6%). In addition, the C16:1 ratio reached a maximum of 25.6% in both OPRs, showing a similar pattern as C16:0 in OPR 1 and gradually decreasing with time in OPR 3. The staple lipid composition for biodiesel production was better in OPR 3 compared to OPR 1, except for C16:1. In addition, we found traces of other fatty acids and no significant difference between OPRs 1 and 3. However, we identified arachidonic acid (C20:4; 0.19%) and eicosapentaenoic acid (C20:5; 1.38%) only in OPR 1 and behenic acid (C22:0; 0.19%) in OPR 3. Also, the mean C16:0, C18:1, and C18:2 composition of the biodiesel was 24.5, 17.0, and 6.1%, respectively, in OPR 1 (47.6% in total) and 24.2, 22.0, and 5.8%, respectively, in OPR 3 (52% in total). The total UFA concentration in the biodiesels derived from OPR 1 and 3 was 65.1 and 67.2%, respectively, approximately twice the saturated fatty acid (SFA) concentrations (34.9 and 32.8%). Therefore, the microalgal biomass produced through mass cultivation, especially OPR 3 with a high C18:1 content, has potential as feedstock for producing biodiesel of the desired quality.

## Biodiesel quality evaluation

The viscosity, ignition quality, oxidative stability, and cold flow of biodiesel are largely affected by the structure of component fatty acid esters (*Arguelles et al., 2018*). Therefore, it is important to develop a microalgal species with a suitable lipid composition for microalgae-based biodiesel production. In this study, we calculated fuel physicochemical properties using empirical equations (based on fatty acid profile analysis) because proximate and ultimate analyses and fatty acid profiling showed that mass-cultured biomass has the potential to be used in biofuel production (Table 1 and Fig. 4). The parameters analyzed included SV, IV, DU, LCSF, CFPP, oxidation stability, kinematic viscosity, and density. Fuel quality properties were also determined using empirical equations and were within the limits of biodiesel standards ASTM D6751 and EN 14214. The biodiesels derived from

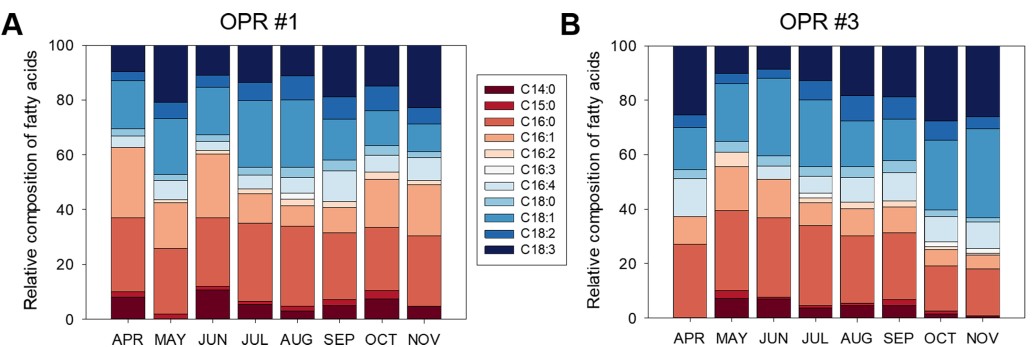

**Figure 4 Fatty acid composition of biomass in OPRs.** Relative fatty acid analysis in (A) OPR 1 and (B) OPR 3. Fatty acid composition was set at 100%. Wine, C14:0; red, C15:0; light-red, C16:0; apricot, C16:1; incarnadine, C16:2; white, C16:3; light sky blue, C16:4; sky blue, C18:0; light-blue, C18:1; blue, C18:2; navy, C18:3. OPR, open pond raceway.

OPRs 1 and 3 had a low density (0.87 g cm$^{-3}$) and low kinematic viscosity (3.95 mm$^2$ s$^{-1}$). In addition, values of the other parameters were as follows: CN, 56.27 and 52.97, respectively; oxidation stability, 13.02 and 11.66 h, respectively; CFPP, $-3.8$ and $-3.0$ °C, respectively; and IV, 85.83 and 97.16 g I$_2$ (100 g)$^{-1}$ fat, respectively. The mean SV, DU, and LCSF were 203.60, 62.75 wt%, and 4.03 wt%, respectively, in OPR 1 and 201.83, 69.37 wt%, and 4.31 wt%, respectively, in OPR 3 (Table 2).

## Possibility of microalgae as biological enhancers

The microalgal biomasses in OPRs 1 and 3 had the potential of feedstock for biodiesel production. Microalgae have been used in various industries, such as biofuel, cosmetics, bioremediation, and biocontrol, so we examined the application of microalgae as bioenhancers in soybean plants (*Ronga et al., 2019*). As shown in Figs. 5A–5P, the microalgal population biodiversity in 2018 was similar to that in 2017 during mass culture, with the dominant species being *Desmosdesmus*. Figure 5Q shows weather changes during the analyzed periods (July 20 to November 21, 2018). The most distinguishable environmental change was a gradual decrease in air temperature. We conducted environmental adaptation assessment at least twice with V2 stage soybean plants grown after seed germination. The microalgae bioenhancer ability was based on the phenotype and seed germination. The first assessment found that the group subjected to the microalgal biomass (1–4 and 1–5), clear solution (1–3), or spray (1–6) of OPR 1 showed improved growth development compared to control groups (1–1 and 1–2), while groups treated with the clear solution (3–3) or spray (3–6) of OPR 3 showed an improved phenotype compared to other groups (Figs. 6A–6E and Fig. S13). In addition, the soybean plant seeds treated with cleared microalgal culture solutions, which included microalgal biomass (1–4 and 1–5), had better quality compared to other groups, resulting in increased pod seed weight (Figs. 6F–6H and Fig. S13). Therefore, the microalgal biomasses and supernatants of mass cultures can be used as feedstock for producing biodiesels and biofertilizers, respectively.

**Table 2** Biodiesel quality analysis of the harvested biomass during mass cultivation.

| | APR | | MAY | | JUN | | JUL | | AUG | | SEP | | OCT | | NOV | | EN 14214 | ASTM D6751 |
|---|---|---|---|---|---|---|---|---|---|---|---|---|---|---|---|---|---|---|
| | O1 | O3 | O1 | O3 | O1 | O3 | O1 | O3 | O1 | O3 | O1 | O3 | O1 | O3 | O1 | O3 | | |
| SV | 205.5 | 202.7 | 203.5 | 205.6 | 205.7 | 202.5 | 202.5 | 201.5 | 200.9 | 203.0 | 202.4 | 202.3 | 204.2 | 199.0 | 204.1 | 198.1 | – | – |
| IV (g $I_2$ 100 g$^{-1}$ fat) | 63.3 | 121.2 | 90.2 | 58.3 | 67.4 | 58.3 | 68.1 | 84.17 | 92.4 | 106.3 | 108.7 | 111.1 | 94.6 | 125.8 | 102.0 | 112.2 | $\leq$ 120 | – |
| CN | 60.8 | 49.2 | 55.2 | 49.2 | 59.6 | 62.6 | 60.2 | 57.5 | 56.0 | 52.1 | 51.8 | 51.4 | 54.1 | 49.2 | 52.5 | 52.6 | $\geq$ 51 | $\geq$ 47 |
| DU (wt %) | 52 | 66 | 67 | 52 | 55 | 53 | 55 | 62 | 69 | 69 | 71 | 74 | 69 | 87 | 64 | 82 | – | – |
| LCSF (wt %) | 4.1 | 4.6 | 3.6 | 5.1 | 3.8 | 4.6 | 4.2 | 4.8 | 4.8 | 4.1 | 4.3 | 4.3 | 3.9 | 3.4 | 3.6 | 3.6 | – | – |
| CFPP (°C) | −3.7 | −2.2 | −5.2 | −0.5 | −4.4 | −2.0 | −3.4 | −1.3 | −1.3 | −3.7 | −2.9 | −3.1 | −4.3 | −5.9 | −5.2 | −5.3 | $\leq$5/ $\leq$−20 | – |
| Oxidation stability (h) | 24.5 | 10.0 | 10.7 | 18.9 | 17.4 | 17.3 | 12.2 | 11.9 | 10.5 | 9.2 | 9.1 | 8.8 | 9.5 | 7.8 | 10.3 | 9.4 | $\geq$ 6 | $\geq$ 3 |
| Kinematic viscosity ($v$) (mm$^2$ s$^{-1}$) | 4.0 | 3.8 | 3.9 | 3.8 | 4.0 | 4.2 | 4.1 | 4.1 | 4.0 | 3.9 | 3.9 | 3.9 | 3.9 | 3.9 | 3.8 | 4.0 | 3.5–5.0 | 1.9–6.0 |
| Density ($\rho$) (g cm$^{-3}$) | 0.87 | 0.88 | 0.88 | 0.88 | 0.87 | 0.87 | 0.87 | 0.88 | 0.88 | 0.88 | 0.88 | 0.88 | 0.88 | 0.88 | 0.88 | 0.88 | 0.86–0.90 | 0.82–0.90 |

**Notes.**
O1, Open pond raceway #1; O3, Open pond raceway #3.

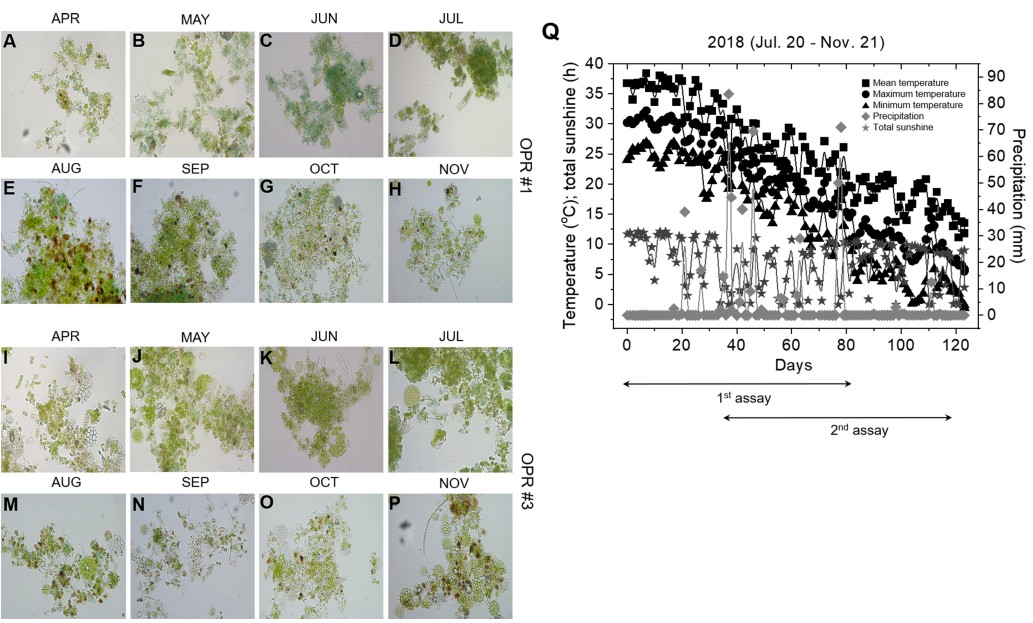

**Figure 5 Microalgal calendar and weather changes during mass cultivation in 2018.** The dominant microalgal species was similar in 2017 and 2018; the presented images were taken under a microscope at 400 × magnification. Microalgal calendar of (A–H) OPR 1 and (I–P) OPR 3 from April to November 2017. (Q) Changes in the mean (square), maximum (circle), and minimum (triangle) temperatures (°C), total sunshine hours (h; star), and precipitation (mm; diamond). OPR, open pond raceway.

# DISCUSSION

## Overview of the MiSeq-based microalgal community during mass cultivation

In previous studies, MiSeq of the 18S rRNA V9 region has failed to determine some species or their abundance in dinoflagellate-diatom mixtures simulated at different ratios (*Guo, Sui & Liu, 2016*). Therefore, various universal primers against 18S rRNA regions have been used to identify microalgal species at the molecular level. Although actin is more suitable as a molecular marker compared to the 18S rRNA gene for analyzing dinoflagellate communities (*Guo, Sui & Liu, 2016*), the 18S rRNA gene is more effective for HTS compared to other candidate genes because 400–500 bp amplicons are required (Fig. S1) (*Ishaq & Wright, 2014*). In addition, various regions of the 18S rRNA gene have been introduced into HTS-based comparative analysis. Of these, the V4 and V8–V9 regions are significant, and taxonomic groups for the V8–V9 region exhibit higher concordance rates compared to the V4 region (Fig. S1). In general, microalgal communities are mostly profiled through morphological identification, followed by qualitative/quantitative assessment using DNA barcoding (*Serrana et al., 2019*). However, this method is not optimal for monitoring microalgal biodiversity in OPRs because of interfering factors, such as the cultivation cycle (month, season, and year) (*Cheung, Allen & Short, 2020*). Metagenomic analysis based on DNA barcoding (the V8–V9 region of the 18S rRNA gene) can be used for rapid, accurate,

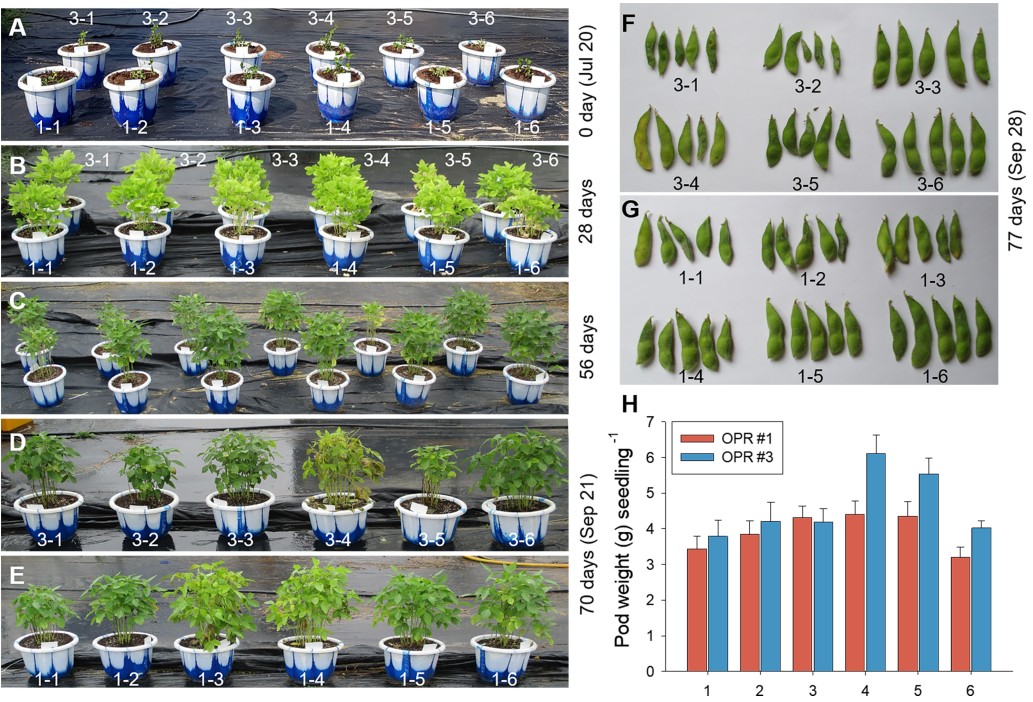

**Figure 6  Biofertilizer assays for soybean plants.** The first biofertilizer experiment was conducted under field conditions from July 20 to September 28, 2018. Soybean seeds were grown in potting soil. V2 seedlings were transplanted after 15 days to 30 cm pots (15 seedlings/pot). The plants were grown for 11 weeks and hand-watered, as needed. Phenotypes in the growth stage were represented at (A) 0, (B) 28, (C) 56, and (D, E) 70 days after soybean seedling transplantation. The effects of a mass culture solution as a soil drench were also examined by (F, G) monitoring seed development and (H) weighing the pods per seedling. a, OPR 1; b, OPR 3; 1–1 and 3–1, soybean plants supplemented with groundwater; 1–2 and 3–2, soybean plants supplemented with eco-sol medium; 1–3 and 3–3, soybean plants supplemented with the clear supernatant from OPRs 1 and 3; 1–4 and 3–4, soybean plants supplemented with microalgal biomasses from OPRs 1 and 3; 1–5 and 3–5, soybean plants supplemented with the cultured solution (supernatant plus microalgal biomasses) from OPRs 1 and 3; 1–6 and 3–6, soybean plants sprayed with the clear supernatant from OPRs 1 and 3; gray bar, OPR 1; black bar, OPR 3. OPR, open pond raceway.

and automatable identification of microalgal species and their precise abundances in microalgal cultures of industrial interest.

## Changes in nutrient parameters during mass cultivation in open pond raceways

With regard to weather changes and nutrient (e.g., nitrogen and phosphorus) consumption, the temperature of OPR 1 was higher compared to OPR 3 during the mass cultivation period except in summer. pH was neutral to heavy alkaline (7.2–10.4) and became increasingly neutral with a gradual increase in water temperature (Fig. 1B). TN and TP consumption was 12.87 mg L$^{-1}$ (86.2%) and 2.26 mg L$^{-1}$(96.3%), respectively, in OPR 1 and 13.14 mg L$^{-1}$ (90.1%) and 2.10 mg L$^{-1}$ (93.5%), respectively, in OPR 3 (Fig. 1C), confirming a higher nutrient removal rate for TP compared to TN. Phosphorus plays an important role in microalgae cellular metabolism by forming many structural and functional components required for growth and development (*Znad et al., 2018*). Naturally occurring mixed

microalgae have been used for wastewater treatment under OPR conditions. *Posadas et al. (2015)* showed in OPR cultivation of *Scenedesmus* sp. that wastewater treatment is confirmed by the effect of the $CO_2$ source. The TN and TP removal rates were $74 \pm 14\%$ and $57 \pm 12\%$, respectively. In another study involving semi-continuous cultivation of *Chlorella* sp., *Scenedesmus* sp., and *Stigeoclonium* sp., wastewater treatment showed a mean TN and TP removal rate of $76.3 \pm 4.0\%$ and $51.5 \pm 3.9\%$ respectively (*Kim et al., 2018*).

## Algal biomass productivity

The DW of OPR 3 was higher compared to OPR 1 during June–August 2017, with the highest difference in summer, when water temperature increased. OPR 3 had a maximum biomass production of 17.6 g m$^{-2}$ d$^{-1}$, while OPR 1 in the greenhouse had a maximum biomass of 13.3 g m$^{-2}$ d$^{-1}$ (Fig. 1E). Under specific environmental conditions of Almería, *Scenedesmus* is consistently the dominant microalgal species in OPRs. In previous studies, biomass productivities ranged from $4 \pm 0$ g m$^{-2}$ d$^{-1}$ in December to $17 \pm 1$ g m$^{-2}$ d$^{-1}$ in July (*Posadas et al., 2015*). The two OPRs had a mean total biomass production of $\sim$8.6–9.9 g m$^{-2}$ d$^{-1}$, corresponding to 0.307–0.309 DW L$^{-1}$ (Figs. 1D–1F). The lower biomass production in OPR 1 was presumably because of the reduced light intensity caused by the greenhouse's semitransparent film, as previously reported (*Giacomelli & Roberts, 1993*). Supplying *Desmodesmus* sp. F51 with inorganic carbon and nitrogen sources at a 1:1 ratio enhanced cell growth and improved biomass production up to 939 mg L$^{-1}$ d$^{-1}$ (*Xie et al., 2017*). Overall, biomass and DW values corresponded to nutrient consumption, highlighting the time when the water temperature was maintained at 20–25 °C, on average, because of the higher atmospheric temperatures from June to September. Microalgae had an optimum growth temperature of 20-−25 °C under normal conditions (*Patel et al., 2019*).

## Microalgal and cyanobacterial communities

Although we found a large number of microalgal species, including *Desmodesmus* sp., *Chlorella* sp., *T. obliquus*, and *P. integrum*, in both OPRs 1 and 3 in a 9 month period, the microalgal species highly dominant in both OPRs was *Desmodesmus* sp. The microalgal abundance of OPR 1 was greater compared to OPR 3. Seasonal changes in light conditions play an important role in sustaining species diversity in OPR systems. The richness of *Desmodesmus* sp. in OPR 3 was 12% higher compared to OPR 1 (Fig. 2). Similarly, in the case of two full-scale wastewater high rate algal ponds (HRAPs) (NHRAPs and SHRAPs) operated over a 23 month cultivation period from spring 2015 to winter 2016 in New Zealand, *Desmodesmus* sp., *Coelastrum* sp., *Micractinium* sp., *Microcystis* sp., and *Pediastrum* sp. were the wastewater microalgal community representative of the dominant species, which changed either seasonally (e.g., with retention time and loading rates) or with species manipulation conditions (e.g., microalgal recycling) (*Sutherland, Turnbull & Craggs, 2017*; *Lutzu & Dunford, 2018*). *Micractinium pusillum* dominated the microalgal community for an 18 month cultivation period in the two HRAP systems (*Sutherland, Turnbull & Craggs, 2017*), while *Desmodesmus* sp. dominated the microalgal community for a 9 month cultivation period despite changes in multiple environmental variables

in the two OPR systems in our study. *Cho et al. (2015)* performed mass cultivation with untreated municipal wastewater in an HRAP for 1 year (2013–2014) in Korea. Observation of the microalgal diversity in semi-continuous mode showed that the dominant microalgal genera in all four seasons were *Scenedesmus*, *Microcystis*, and *Chlorella*. In contrast to our results, *Scenedesmus* sp. persisted in *Cho et al. (2015)* culture system in all four seasons, but the three strains were not co-dominant. To identify the most promising species that can be used as biodiesel feedstock for a large-scale cultivation system in China. *Xia et al. (2014)* cultivated *Desmodesmus* sp. and *Scenedesmus* sp. under outdoor conditions. In waste water treatment coupled with an HRAP, the *Scenedesmus* sp. population, which was the dominant strain in summer (initial stages of cultivation), decreased significantly and *Microcystis* sp. became the dominant species, although it was absent in the preceding months. In spring and summer, *Chlorella* was the dominant genus (*Cho et al., 2015*). The change in dominance from one genus to another was drastic and rapid, typically within 7–10 days (Fig. 2). The turnover rate of dominant species in natural systems may be equally fast when the steady-state phase beaks down because of significant shifts in environmental changes, although the low sampling frequency of many studies has made this difficult to quantify (*Sutherland, Turnbull & Craggs, 2017*). In the HRAP-dependent wastewater treatment process, dynamics of the *Scenedesmus* population are highly correlated with temperature, while those of the *Chlorella* population are highly correlated with the effluent's organic content (*Cho et al., 2015*). Therefore, monitoring the rapid turnover of the dominant species, as shown in our study, might make it challenging to maintain a desirable species at the full-scale.

Microalgal diversity and biomass are likely affected by different parameters, such as environmental factors (e.g., temperature) and to biological and physiochemical parameters (e.g., influent microbial diversity and organic carbon, nitrates, pH, dissolved oxygen, and conductivity levels) (*Zongo et al., 2019*). However, the most significant factor affecting species abundance is temperature. Variations in the local environment (i.e., in the temperature of the immediate surroundings of a community) are crucial, not only for the ecosystem process rate, but also for species assembly persistence and the direct relationship between biodiversity and ecosystem functioning.

In the OPRs 1 and 3 systems, we found no differences in biomass productivity (mean: 8.6–9.9 g m$^{-2}$ d$^{-1}$) and nutrient consumption rates of TN and TP (Figs. 1C and 1F). Accordingly, in microalgal communities in the two full-scale wastewater HRAPs, ~33 microalgal species were found but species abundance was low and not related to either productivity or nutrient removal efficiency (*Sutherland, Turnbull & Craggs, 2017*). As shown in our study, both nutrient removal and biomass production did not differ between the two HRAPs when the dominant species was the same or different between the two ponds (*Sutherland, Turnbull & Craggs, 2017*). These findings indicate that compared to individual species, microalgal communities are more critical for OPR performance. In addition, microalgae mass cultivation for the producing biomass and associated valuable compounds have gained increasing interest not only within the scientific community but also at the industrial level.

In our study, we identified ~26 cyanobacteria species (Figs. S11 and S12), of which *Cyanobium gracile* frequently appeared in both OPRs 1 and 3 in spring and summer, while *Foliisarcina bertiogensis*, *Spirulina major*, *Cyanobacterium aponinum*, and *Microcystis aeruginosa* bloomed in the fall. However, their populations were higher in OPR 1 compared to OPR 3. *Synechococcus elongatus* and *Calothrix desertica* were identified only in OPR 3 in the fall (Fig. S12). In the two full-scale wastewater HRAPs, identified cyanobacteria, such as *Dolichospermum circinale*, *Merismopedia tenuissima*, *M. aeruginosa*, *Microcystis flosaquae*, *Oscillatoria* sp., and *Planktothrix isothrix*, with the dominant species being *M. aeruginosa* (*Sutherland, Turnbull & Craggs, 2017*). The high richness of *M. aeruginosa* for a short duration resulted in a change in the dominant species (*Sutherland, Turnbull & Craggs, 2017*). In addition to the microalgal community, nutrient concentrations of nitrogen and phosphorus and organic carbon loads affect cyanobacteria dominance (*Cho et al., 2015*).

Under harsh conditions, *Desmodesmus* sp. produces a higher amount starch grains, oil bodies, and cell wall polysaccharides and increases its symbiotic interactions with taxonomically distant invertebrates (e.g., hydroids, sponges, and polychaetes) (*Baulina et al., 2016*). These symbiotic relationships include all possible interactions known in nature, such as mutualism, commensalism, and parasitism (*McKay, Gibbs & Vaughn, 1991*). Cyanobacteria can also have symbiotic interactions with microalgae. For example, the co-culture of the Louisiana-native microalga *Chlorella vulgaris* and the cyanobacterium *Leptolyngbya* sp. shows commensalism to produce lipids for biodiesel production (*Subashchandrabose et al., 2011*). On the basic of the symbiotic relationship, microalgae cyanobacteria consortia can lead to a high microalgal growth rate, improved nutrient and pollutant uptake, and production of metabolites with high biotechnological application potential (*Silaban et al., 2014*). The exact metabolic mechanisms underlying such interactions are unclear. However, the benefits of this growth strategy can potentially be exploited in different biotechnology fields, such as biofuel production, especially if the microalgal species involved possesses a high lipid contents (*Lutzu & Dunford, 2018*).

## Characteristics of microalgal biomass, and fatty acids in mass cultivation

The microalgal community is a more important determinant of freshwater systems and microalgal biomass production in OPRs 1 and 3 compared to one individual species. The mean weight percentage CHNSO values were 43.22, 6.23, 6.23, 0.46, and 30.31, respectively, in OPR 1 and 41.50, 5.86, 5.62, 0.41, and 26.77, respectively, in OPR 3. The mean weight percentage CHNSO values of *Hapalosiphon* sp. biomass are 47.94, 7.44, 6.45, 0.58, and 37.58, respectively, while those of *Botryococcus braunii* biomass are 77.04, 12.40, 1.23, 0.18, and 9.86, respectively (*Liu et al., 2012*). In our study, the CVs were higher in summer and the fall compared to spring and were higher in OPR 1 compared to OPR 3 (Table 1). The mean CVs of OPR 1 and 3 were 19.5 and 18.5 MJ kg$^{-1}$, respectively, which are higher compared to the land plant biomass (18.4 MJ kg$^{-1}$) (*Ross et al., 2008*). In general, microalgae grown under normal conditions have a CV of 18–21 MJ kg$^{-1}$ (*Scragg et al., 2002*). The CV of *Hapalosiphon* sp. is 14.75 MJ kg$^{-1}$, similar to that of sewage sludge or lignocellulose biomass (19–25 MJ kg$^{-1}$), while the CV of *B. braunii* is as high as

35.58 MJ kg$^{-1}$ because of total carbon and hydrogen contents of 89.4% in the microalgal biomass, similar to that of crude oil (*Liu et al., 2012*). Considering a high carbon and hydrogen content an advantage for feedback in biofuel production, our results showed that the microalgal biomasses in OPRs 1 and 3 were feasible for biodiesel production, although the produced values were lower compared to *B. braunii*.

The major fatty acids of the microalgal biomasses in OPRs 1 and 3 were C16:0 (24.5 and 24.2%, respectively), C16:1 (15.4 and 9.7%, respectively), C16:4 (6.0 and 7.6%, respectively), C18:1 (16.9 and 21.9%, respectively), C18:2 (6.1 and 5.8%, respectively), and C18:3 (14.1 and 17.9%, respectively) (Fig. 4). The fatty acids produced by microalgae are C16:0, C18:0, C18:1, C18:2, and C18:3 (*Huang et al., 2015*), of which UFAs (C18:1 and C18:2) are important components of high-quality biodiesel (*Bouaid, Martinez & Aracil, 2007*). In our study, mean C16:0, C18:1, and C18:3 content of the biodiesel was 24.5, 17.0, and 14.5%, respectively, in OPR 1 (56.0% in total) and 24.2, 22.0, and 17.9%, respectively, in OPR 3 (64.1% in total) (Fig. 4). Especially, C18:1 is an important indicator of biodiesel quality and provides great stability (*Ho et al., 2014*). The total UFA concentration in the biodiesels derived from OPRs 1 and 3 was 65.1 and 67.2%, respectively, which was approximately two times greater compared to SFAs (34.9 and 32.8%, respectively). The low PUFA content and high SFA and MUFA content are among the most important indicators of biodiesel quality (*Piligaev et al., 2018*). The fatty acid profiles of microalgal oils generally affect the qualities of the biodiesel produced. High SFA and MUFA contents lead to high oxidative and thermal stabilities, resulting in a slower deterioration rate of lipid characteristics (*Mostafa & El-Gendy, 2017*).

In our study, we found high levels of fatty acids with a high degree of SFAs, such as C16:0. In addition, MUFAs C16:1 and C18:1 were abundant in OPRs 1 and 3 (Fig. 4). According to biodiesel standard EN 14214, the percentages of linolenic acid (C18:3) and PUFAs (P4 double bond) should not be greater than 12 and 1%, respectively (*Mandotra et al., 2014*). The fatty acid profiles of microalgal biomasses in OPRs 1 and 3 had a high linolenic acid content of 14.2 and 18.0%, respectively. Assessment of the amount of FAMEs with ≥ 4 double bonds for microalgal crude oil extract has shown that the microalgal strain does not produce, FAMEs and therefore represents a good biodiesel source (*Mandotra et al., 2014*). Microalgal fatty acid profiling of biodiesel obtained from *Desmodesmus* sp. I-AU1 contains a total SFA methyl ester content of 31.0%, while the total MUFA C18:1 content is 25.6% (*Arguelles et al., 2018*). The SFA and MUFA percentage composition of the microalga was 56.7%, which is high relative to related studies (*Arguelles et al., 2018*). The fatty acid content of microalgal biomasses in OPRs 1 and 3 mainly comprised SFAs and MUFAs (67.3 and 64.8%, respectively), which are suitable for biodiesel production.

## Properties of microalgal biomass-based biodiesel quality parameters in mass cultivation

The fatty acid profile of microalgal oils generally affects biodiesel quality (*Nascimento et al., 2013*). The biodiesels derived from OPRs 1 and 3 had a low density (0.87 g cm$^{-3}$) and low kinematic viscosity (3.95 mm$^2$ s$^{-1}$). In addition, values of the other parameters were as follows: CN, 56.27 and 52.97, respectively; oxidation stability, 13.02 and 11.66

h, respectively; CFPP, $-3.8$ and $-3.0$ °C, respectively; and IV, 85.83 and 97.16 g $I_2$ (100 g)$^{-1}$fat, respectively. The mean SV, DU, and LCSF were 203.60, 62.75 wt%, and 4.03 wt%, respectively, in OPR 1 and 201.83, 69.37 wt%, and 4.31 wt%, respectively, in OPR 3 (Table 2). The carbon chain lengths of both SFAs and UFAs in crude oil extracts generally affect biodiesel properties, such as CN, oxidative stability, and cold–flow properties (*Samorì et al., 2013*). Generally, crude oil extracts with a high proportion of SFAs and MUFAs are preferred for biodiesel production because these fatty acids increase the energy yield and oxidative stability of biodiesel. On the one hand, oils containing MUFAs are prone to solidification at low temperature; on the other hand, oils rich in PUFAs have good cold-flow properties but are vulnerable to oxidation. This has an adverse effect on fuel conservation and combustion (*Arguelles et al., 2018*). The total mean percentages of SFAs and MUFAs and of SFAs and PUFAs for the total biomass were 67.3 and 64.7%, respectively, and 67.6 and 68.4%, respectively (Fig. 4). With regard to biodiesel quality, microalgal biomasses in OPRs 1 and 3 showed good fuel viscosity, equal or lower density, higher CN and oxidation stability, and good performances at low temperature. Overall, our results indicate that biodiesel produced from the oil *Desmodesmus* sp. grown in open ponds (OPR 1 or 3) will have low emissions, good combustion, good flow properties at low temperatures, and excellent oxidation stability.

### Application of microalgae as biological enhancers

The bioenhancing ability of microalgae is based on the phenotype. The first assessment showed that the group subjected to the microalgal biomass (1–4 and 1–5), cleared solution (1–3), or spray (1–6) of OPR 1 has improved growth compared to control groups (1–1 and 1–2), while groups treated with the cleared solution (3–3) or spray (3–6) of OPR 3 display an improved phenotype compared to other groups (Figs. 6A–6E and S13). Consistent with our results, microalgae have been considered a potential biofertilizer to replace mainstream synthetic fertilizers in order to enhance the biological and chemical properties of soil, positively affecting plant characteristics and increasing yields (*Ronga et al., 2019*). For example, inoculation of *C. vulgaris* and *S. dimorphus* increases rice yields by improving nitrogen fixation (*Garcia-Gonzalez & Sommerfeld, 2016*). The culture solution, cellular extracts, and dry biomass of *Acutodesmus dimorphus* enhances seed germination, plant growth, and floral production in Roma tomato (*Jochum, Moncayo & Jo, 2018*). Therefore, the microalgal biomass and culture solutions of OPRs 1 and 3 improved the agronomic traits of soybean plants and can be introduced as biological enhancers, such as soybean spray and biofertilizer.

### CONCLUSIONS

This study investigated the species diversity of the microalgal community during mass cultivation. *Desmodesmus* sp. was found to be the dominant microalgal species throughout the evaluation period. The microalgal biomasses produced from OPRs 1 and 3 are rich in SFAs (C16:0), MUFAs (C16:1 and C18:1), and PUFAs (C16:4 and C18:3). The biodiesel derived from the microalgal biomasses showed good fuel viscosity, equal or lower density, higher CN and oxidation stability, and good performance at low temperature, meeting

the major biodiesel standards. The microalgal biomass and culture solution of OPRs 1 or 3 improved the environmental adaptation of soybean plants. Therefore, OPR-grown *Desmodesmus* sp. is a potential alternative feedstock for producing excellent-quality biodiesel and biofertilizer. In addition, the understanding of microalgal diversity can help sustain a dominant genus in OPR systems, maximizing overall efficiency. Following a decrease in water temperature, eukaryotic diversity is highest in summer compared to spring and the fall. In addition, quantitative analysis of the phytoplankton community, which is important for ecological studies, has problems of low accuracy of species identification (which heavily depends on personnel experience) and low efficiency and is also time consuming. As an alternative to overcome such limitations, our results could be applied to develop an improved technique for accurate and fast quantification of microalgal samples via HTS of barcoding genes.

### Funding
This study was supported by a grant from the National Research Foundation of Korea (NRF-2017R1A2B4002016). This work was supported by a grant from the Next-Generation BioGreen 21 Program (No. PJ013240), Rural Development Administration, Korea. The funders had no role in study design, data collection and analysis, decision to publish, or preparation of the manuscript.

### Grant Disclosures
The following grant information was disclosed by the authors:
National Research Foundation of Korea: NRF-2017R1A2B4002016.
Next-Generation BioGreen 21 Program, Rural Development Administration, Korea: PJ013240.

### Competing Interests
The authors declare there are no competing interests.

### Author Contributions

- Seung-Woo Jo conceived and designed the experiments, performed the experiments, analyzed the data, prepared figures and/or tables, authored or reviewed drafts of the paper, and approved the final draft.
- Jeong-Mi Do performed the experiments, analyzed the data, authored or reviewed drafts of the paper, and approved the final draft.
- Ho Na performed the experiments, prepared figures and/or tables, and approved the final draft.
- Ji Won Hong analyzed the data, authored or reviewed drafts of the paper, and approved the final draft.
- Il-Sup Kim conceived and designed the experiments, analyzed the data, prepared figures and/or tables, authored or reviewed drafts of the paper, and approved the final draft.

- Ho-Sung Yoon conceived and designed the experiments, authored or reviewed drafts of the paper, and approved the final draft.

## Data Availability

The raw measurements are available in Tables S2 and S3.

The sequences are available at GenBank: *Scenedesmus* sp. KNUA019 (MT644350); *Chlamydomonas* sp. KNUA021 (JN863299); *Desmodesmus* sp. KNUA024 (MT603634); *Chlorella vulgaris* KNUA027 (KU306723); *Acutodesmus* sp. KNUA038 (KT883908); *Pseudopediastrum* sp. KNUA039 (KT883909); *Chlamydomonas* sp. KNUA040 (KY655002).

## Supplemental Information

Supplemental information for this article can be found online at http://dx.doi.org/10.7717/peerj.9418#supplemental-information.

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
