# Peer review of "Assessment of biomass potentials of microalgal communities in open pond raceways using mass cultivation"

_PeerJ, doi:10.7717/peerj.9418_

## Round 0.1 · original submission · Major Revisions

The following issues have been identified which need attention:
[1] Authors should comment on the issue of amplification bias in the metagenomics analysis, and how this might influence their results;
[2] The authors should consider expanding their discussion of the results, and not simply report what was found. Placing the work in the context of the area of application, and in relation to current knowledge is an essential step in scientific reporting.

·

Basic reporting

1. The error bars may be lost in Figure 1 D) and H). Please add into these figures.
2. Some key fatty acids (i.e. EPA-C22, DHA-C24) may be lost in Figure 4. According to the microscopic photo in Figure 3 and 5, I think there are diatoms which are rich in polyunsaturated fatty acids, particularly DHA. However, there is no fatty acids with carbon >20 in the algal communities (Fig.4). Please check carefully the results of fatty acids composition, and revise Figure 4.

Experimental design

1. The part, applicability of microalgae a biological enhancer for soybeans, the two applications (biodiesel production and biofertilizer assay) of mass cultivation of microalgae have contradictions each other. I think it is priority to retain lipid (biodiesel) production rather than biofertilizer aspect.
2. The biofertilizer assay is also a good filed for the application of mass cultivation of microalgae, but I think it is better to have an independent and deep study.

Validity of the findings

See General comments for the author.

Additional comments

General comments:
The extent to which amplification bias affects the accuracy of microalgal community analysis is currently unknown. This study integrates a large-scale outdoor growth model with biodiversity data from a microalgal community in 2017 to estimate the long-term lipid and biomass production potential of microalgae cultured in a mass cultivation system in Korea. In addition, the application of outdoor mass cultivated microalgae to soybean plants was investigated. This paper is very rich in experimental contents, including microalgal cultivation, the analysis of genes sequencing, biodiversity, environmental parameters and nutrients, assessment of lipids content and quality, and application aspects. It is of great importance to evaluate the potential of outdoor mass cultivation of microalgae on energy production and the fertilizer industry. However, the presentation is not compact enough due to dispersed contents. Some spelling mistakes are found. The detailed comments are as follows:

Abstract:
Line 39: the biodiversity analysis of mass cultivation systems is based on high throughput sequence (HTS) in this paper, while “using metagenomics” here is not accurate.
Line 48: The seasonal changes and diversity of other organisms are the main factors affect mass cultivation of microalgae. More detailed results should be given on how these factors affect the systems. And, what are the other organisms? Other microalgae except for Desmodesums sp. or cyanobacteria and others? Line 46-48, the sentence is unclear.
Line 51-52: It is suggested to give the important data that can directly show OPR #1 enhanced the overall lipid yields.
Introduction
The whole introduction part seems to be not focused on the key points. In this part, the authors emphasize on sequencing technology, especially the primers selectivity in HTS. However, the main points in this study are effect of biodiversity in different systems on microalgal cultivation, energy production and its application. As far as l know, the THS technology is worldwide used and very common and the primers V4, V8-V9 the authors mentioned are reported previously. This paper is totally different from the paper by Bradley 2016 the authors cited. Thus, it is advised to reduce some contents on sequencing technology and give more information on factors affecting outdoor mass cultivation of microalgae.
Line 94: this cited paper was published in 2012, which was not recently published.
Line 115-116: “it had identified that conserved regions that may be best suited for amplifying hypervariable regions”. I am confused about this conclusion. The conserved regions or the conserved regions close to hypervariable regions?
Line 126-126: The V4 primer is commonly used in amplifying eukaryotic organisms. However, previous study by Bradley had been found V8-V9 was more suitable for amplifying eukaryotic organisms. The authors then use this method in the present study. Thus, these descriptions here need to be modified.

Material and methods
Line 138: here semi-continuous cultivation was carried out, while in line 153, Continuous cultivation was carried out.
Why use the groundwater with Eco-sol medium rather than domestic or unban wastewater. They contain similar nutrients level.
Line 146: CO2 was injected at 10 L min-1. What is the CO2 concentration?
Line 165-188: Miseq-based microalgae diversity, this part is too complicated. It is highly recommended to conclude the information in a table where specific primers are listed for amplification of microorganisms.
Line 179: the conserved region of V8-V9 was used for PCR amplicon?
Line 190: mock community construction. This part seems not important in this paper. As I found, this part was similar to the published paper by Bradley 2016.
Biofertilizer assays: this paper tries to evaluate the potential of outdoor mass cultivation of microalgae on energy production and the fertilizer industry. Although both aspects are showed better applicability, it seem that these two applications of mass cultivation of microalgae have contradictions each other. It is priority to retain lipid production rather than biofertilizer aspect. The biofertilizer assay is also a good filed for the application of mass cultivation of microalgae, but it is better to have an independent and deep study.
Line 243, 263-269: the fonts of equations need to be uniformed with the contents.

Results and Discussion
The structure of this part is not compact enough due to dispersed contents and results description is not concise, e.g. line 342-345,396-408. The four main contents: (a) Qualitative significance verification of HTS analysis, (b) Microalgal biodiversity of MiSeq analysis, (c) Microalgal succession during mass cultivation, (d) Microalgal and cyanobacterial compositions during mass cultivation combine as biodiversity of mass cultivation should be better.
Line 326: the prokaryotic community was analyzed using a well-known primer (specific primer needs to give here) set for 16S rRNA. The percentage of other bacteria except cyanobacteria is nor displayed. Do the bacteria have some influence on mass cultivation of microalgae?
Line 358-368: The discussions here are not closely related to the former results.
Line 365-371: this part introduces biomass and lipid productivity, and calorific value, which is again displayed later. It is better to remove this part.
Line 444-445: what are r-strategists?
Line 474-479: did nutrients starvation happen in the mass cultivation? What is the purpose of discussing on nutrients starvation?
Some spelling mistakes are found, e.g. line 433, 454, and 499.

Reviewer 2 ·

Basic reporting

The authors used the metagenomics technology to combine the large-scale outdoor growth model with the biodiversity data of microalgae community, trying to estimate the lipid content and biomass production potential of microalgae in the open ponds. The data has been enough. But there are still some apparent flaws in the paper. Therefore, the paper is considered to be accepted after careful revision. The commons are as follows:
1. I suggest that you improve the description at lines 46-54, these sentences are complicated. Please simplify the language.
2. Rewrite the introduction. Your introduction needs more detail. The main content in INTRODUCTION is the method used in this study. So, if the authors wanted to report the findings from the view of “the new method”, please consider revising the title of this paper. Besides, please consider providing the related results of the traditional method to the algal community in the authors’ cultivation systems in order to highlight the accuracy and convenience of the new method.
But, if the authors wanted to report the findings from the perspective of the “Algal Community”, in the INTRODUCTION,they need to state the current findings about the algal community in the open ponds and the things needed to research in the future, to emphasize the novelty of the findings in this paper.
3. In the whole text, please organize by importance and the highlights of the issues.
4. In “Results and discussion”, It seems only to be the result description. Please consider digging the deeper reasons, such as, the relationships between the changes of the various environmental conditions and the algal community, algal lipid content, algal biodiesel quality and so on. There are not enough references for the analysis of the results. Especially for the discussion of each index result.
5. Please consider comparing the results in this paper with those in others.
6. In lines 358-364, the analysis of the reference can not be well compared to your study.

Experimental design

1. The authors seemed not to clearly describe the purpose of this experiment. Please consider explaining why the authors did the research about the algal community in the open ponds.
2. For the “Biofertilizer Assays”, please consider stating the aim of each experimental setup.

Validity of the findings

NO COMMENTS.

---

## Round 0.2 · Minor Revisions

While your manuscript has been deemed acceptable for publication, we strongly recommend having a native English speaker aid in revising the text.

·

Basic reporting

I feel that the abstract can be more concise.
Focus on the important results and choose the points you want to emphasize.

Experimental design

No comment.

Validity of the findings

No comment.

Additional comments

This manuscript has been improved.
I agree to publish the manuscript.

Reviewer 2 ·

Basic reporting

All the revisions have been well done. But the article needs to be revised by native English speakers before it can be published

Experimental design

no comment

Validity of the findings

no comment

Additional comments

All the revisions have been well done. But the article needs to be revised by native English speakers before it can be published

---

## Round 0.3 · accepted · Accept

Thank you for all your efforts in revising this paper, which is now suitable for publication in PeerJ.